# Health engagement: a systematic review of tools modifiable for use with vulnerable pregnant women

Jyai Allen ,[1] Debra K Creedy ,[1] Kyly Mills,[2] Jenny Gamble [3]

¹School of Nursing and Midwifery, Griffith University, Nathan, Queensland, Australia
²School of Health Sciences and Social Work, Griffith University, Gold Coast, Queensland, Australia
³School of Nursing, Midwifery and Health, Coventry University Faculty of Health and Life Sciences, Coventry, UK

**Correspondence to**
Professor Jenny Gamble;
ad7603@coventry.ac.uk

## ABSTRACT

**Objective** To examine available health engagement tools suitable to, or modifiable for, vulnerable pregnant populations.

**Design** Systematic review.

**Eligibility criteria** Original studies of tool development and validation related to health engagement, with abstract available in English, published between 2000 and 2022, sampling people receiving outpatient healthcare including pregnant women.

**Data sources** CINAHL Complete, Medline, EMBASE and PubMed were searched in April 2022.

**Risk of bias** Study quality was independently assessed by two reviewers using an adapted COSMIN risk of bias quality appraisal checklist. Tools were also mapped against the Synergistic Health Engagement model, which centres on women's buy-in to maternity care.

**Included studies** Nineteen studies were included from Canada, Germany, Italy, the Netherlands, Sweden, the UK and the USA. Four tools were used with pregnant populations, two tools with vulnerable non-pregnant populations, six tools measured patient–provider relationship, four measured patient activation, and three tools measured both relationship and activation.

**Results** Tools that measured engagement in maternity care assessed some of the following constructs: communication or information sharing, woman-centred care, health guidance, shared decision-making, sufficient time, availability, provider attributes, discriminatory or respectful care. None of the maternity engagement tools assessed the key construct of buy-in. While non-maternity health engagement tools measured some elements of buy-in (self-care, feeling hopeful about treatment), other elements (disclosing risks to healthcare providers and acting on health advice), which are significant for vulnerable populations, were rarely measured.

**Conclusions and implications** Health engagement is hypothesised as the mechanism by which midwifery-led care reduces the risk of perinatal morbidity for vulnerable women. To test this hypothesis, a new assessment tool is required that addresses all the relevant constructs of the Synergistic Health Engagement model, developed for and psychometrically assessed in the target group.

**PROSPERO registration number** CRD42020214102.

## INTRODUCTION

Health engagement is the willingness and ability of people to actively participate in their

### STRENGTHS AND LIMITATIONS OF THIS STUDY

⇒ Health engagement in maternity care is an emerging concept, and terms used in this review may not have captured novel or less-used terms.
⇒ Study quality was independently assessed by two reviewers using a quality appraisal checklist.
⇒ The theoretical model used to synthesise results was missing two constructs relevant to vulnerable populations: health literacy and cultural safety.

health and navigate healthcare systems.[1] This definition includes two related but separate constructs in relation to health and healthcare: activation and health literacy.[2] Activation includes the motivation and confidence to autonomously manage one's health and healthcare, and has been associated with positive health outcomes in several populations with chronic conditions.[3] Health literacy is the extent to which individuals can 'obtain, process and understand basic health information',[4] which impacts their behaviours and outcomes.[5] In psychology and related disciplines, the concept of engagement is often measured by therapeutic alliance between patient/client and provider. Therapeutic alliance refers to a collaborative and emotional bond between patient/client and provider that predicts better psychological outcomes with greater certainty than specific interventions.[6]

A recent systematic review demonstrated that therapeutic alliance is rarely measured in nursing, most commonly in mental health nursing (eight papers).[7] A preliminary literature search by the authors found it has never been measured in midwifery, and there has been only one qualitative study of therapeutic alliance in the context of maternity care.[8] Several systematic reviews demonstrate that engagement interventions can improve patients' knowledge and experience, use of health service, health behaviour and health outcomes.[9] However, there are few theoretical frameworks available to inform research

about patient engagement, alongside a lack of validated and specific measurement tools.[10] Furthermore, the extent to which such tools are suitable for vulnerable pregnant populations is under-researched.

## Vulnerable pregnant women

The term *vulnerable women* encapsulates those women most at risk of poorer maternal health outcomes. In Australia, these populations include adolescent mothers; Aboriginal and Torres Strait Islander mothers; culturally and linguistically diverse mothers; mothers from lower socioeconomic backgrounds; and mothers who reside in regional and remote areas. Vulnerable women are more likely to have socially determined risk factors which independently predict poor pregnancy outcomes like low birth weight and/or preterm birth.[11] These factors have been systematically researched and include depression,[12] intimate partner violence,[13] housing instability,[14] environmental chemical exposure including tobacco,[15] illicit drug use,[16] inadequate or excessive gestational weight gain,[17] genital tract infection,[18] and late or inadequate use of antenatal care.[19] Importantly, the aforementioned factors are modifiable through access to, and engagement with, high-quality maternity care.

In order to achieve optimal benefits, maternity care models need to do more than enable pregnant women to just *turn up* to their maternity appointments; the system needs to facilitate *buy-in*.[20] People buy-in when they make 'emotional investment and commitment' to healthcare because they are enabled and believe it is 'worthwhile and beneficial'.[21] People demonstrate buy-in when they participate in health-promoting behaviours (eg, attend for care, have screening tests), and participate in self-care activities including nutrition and exercise while minimising harm (eg, reducing or ceasing substance use).[3] Pregnant women buy-in because they hope and believe that this will improve the health and well-being of themselves and their babies.[11] Furthermore, patient buy-in is predicated on trust in health providers' advice and guidance.[22]

There is a clear need to measure health engagement to better understand and address the poorer maternal health outcomes of pregnant women from vulnerable groups. However, we were unable to locate any reviews of health engagement tools for use with pregnant women. Many available health engagement tools have been developed with inpatients who have chronic health conditions, which may not translate to healthy pregnant women receiving outpatient primary care. This gap highlights the need to critically review existing tools to determine their applicability for use with vulnerable pregnant women.

## Objectives

This review aimed to (1) describe the nature of health engagement tools used in outpatient populations with or without chronic health conditions; (2) evaluate the reliability and validity of those tools; and (3) use a health engagement framework to evaluate which tools,

or components of tools, could be modified for use with vulnerable pregnant populations.

## METHODS

The systematic review protocol was registered with PROSPERO (number CRD42020214102).

## Theoretical framework

This review was informed by elements of Synergistic Health Engagement (SHE)—an empirical model based on integrated findings from mixed-methods research with pregnant adolescents accessing caseload midwifery.[11] Caseload midwifery provides continuity through one-to-one, relationship-based care during pregnancy, birth and until the first 6 weeks.[23] According to the SHE model, the first three constructs create the conditions for *buy-in to maternity care* (the fourth construct), which modifies *socially determined predictors of perinatal morbidity* (the fifth construct). The five constructs and their elements are listed below:

1. *Optimal model of care:* continuity of midwifery carer, with 24-hour telephone availability, with community-based, flexible visits.
2. *Midwife's attributes:* the skills and personal qualities that enable her to be empowering, trustworthy and empathetic.
3. *Philosophy/best practice:* the midwife's use of health promotion, woman-centred care, shared decision-making.
4. *Woman's buy-in:* having hope and belief that maternity care will be beneficial, disclosing risk factors, trusting and acting on health advice, and engaging in self-care.
5. *Modifiable social risks for perinatal morbidity:* inadequate antenatal care; smoking, alcohol or drug use; poor mental health; family violence; inappropriate gestational weight gain and genitourinary infection.[20]

The SHE model explains the mechanism by which caseload midwifery reduces the likelihood of poor pregnancy outcomes for vulnerable women.[20] Therefore, SHE was a suitable framework to assess the construct of tools to measure health engagement in maternity care. Application of the SHE model informed the synthesis of items included in the tools and the interpretation of findings.

## Eligibility criteria
### Inclusion criteria

Primary research with study populations of adolescents or adults, and used structured tools to assess healthcare engagement or a related construct (including patient activation, patient–professional relationship).

### Exclusion criteria

Primary research population inpatients, children or the elderly (mean age 65 years or older); engagement not related to receiving healthcare (eg, technology, research, policy, patient safety); participation not related to healthcare (eg, work or community participation); ineligible study design (pilot studies, intervention studies,

**Table 1** Search strategy

| Databases | Search terms | Filters |
|---|---|---|
| Cumulative Index of Nursing and Allied Health Literature Complete | patient activation OR patient empowerment OR patient engagement OR patient involvement OR patient participation | Publication type—questionnaire/scale |
| Medline and EMBASE | (patient activation OR patient empowerment OR patient engagement OR patient involvement OR patient participation) AND (tool OR index OR measure OR scale OR instrument OR questionnaire) | Date range (2000–2022) Peer reviewed Age groups 13–44 years |
| PubMed | Medical Subject Heading (MeSH) term *patient participation* AND MeSH major heading *surveys and questionnaires* | |

qualitative studies, case studies, systematic reviews); earlier iterations of tools which had since been revised or validation of translated tools; shared decision-making tools; tools focused on a specific health condition.

### Search strategy and information sources
Table 1 details the search that was conducted in April 2022.

### Selection and data collection process
We reviewed articles identified by these searches and relevant references found within those articles. When we identified multiple eligible papers from the same study, we included the main paper reporting validity and reliability of the tool. For every eligible study, two reviewers extracted data on study design, setting, sample size, population type, age range, research design, name of tool and psychometric properties of the tool.

### Study risk of bias assessment
To our knowledge, there is no suitable tool for assessment of health engagement tools for this population. Therefore, two reviewers independently assessed the quality of included studies using a study-specific tool. We adapted the COSMIN Risk of Bias checklist and updated criteria for measurement properties.[24] Our 10 items for measurement properties included: clear purpose, theoretical constructs, content validity, pilot test, structural validity, internal consistency, reliability, as well as criterion, convergent and predictive validity. Each item was scored according to 1 = adequate or 0 = doubtful/inadequate. Scores were summed with scores of 9–11 = high quality; 5–8 = moderate quality and <5 = poor quality. A third reviewer was consulted if disagreements occurred, and a majority decision was reached. We investigated the theoretical basis of included tools and compared identified constructs with elements of the SHE model to evaluate utility of tools for use with vulnerable pregnant populations.

### Patient and public involvement
None.

## RESULTS
### Study selection
Figure 1 reports the number of studies obtained, screened, and reviewed through databases and other methods. The database search strategy resulted in two authors reading 41 full-text articles to further determine eligibility and resolve any disagreement about inclusion by discussion and consensus. A third reviewer was consulted if disagreements occurred, and a majority decision was reached. Systematic reviews were screened, and where indicated, read in full to search for additional tools, but systematic reviews were not themselves included in this study. Figure 1 indicates reasons for excluding articles from the systematic review following close reading. The authors were aware of three relevant tools that met the eligibility criteria but were not located through the search of databases. A website search for these three tools was conducted; they were located and included in the review.

### Study characteristics
Nineteen tools related to health engagement from 19 studies published between 2000 and 2022 were included. Table 2 summarises the key characteristics of the 19 tools included in the review. Studies were conducted in Canada,[25–28] Germany,[29 30] Italy,[31 32] the Netherlands,[33] Sweden,[34] the UK[35 36] and the USA.[37–43] Participants were men and women who were 11 years or older; mean age of participants in each of the included studies was <65 years of age. Four tools were tested with the pregnant population.[25–27 43] Of these, two tools were co-designed with vulnerable pregnant women (eg, refugee, socially disadvantaged),[25 27] and one study adapted a pre-existing tool (Interpersonal Care Processes) to suit a pregnant population (Prenatal Interpersonal Care Processes).[43]

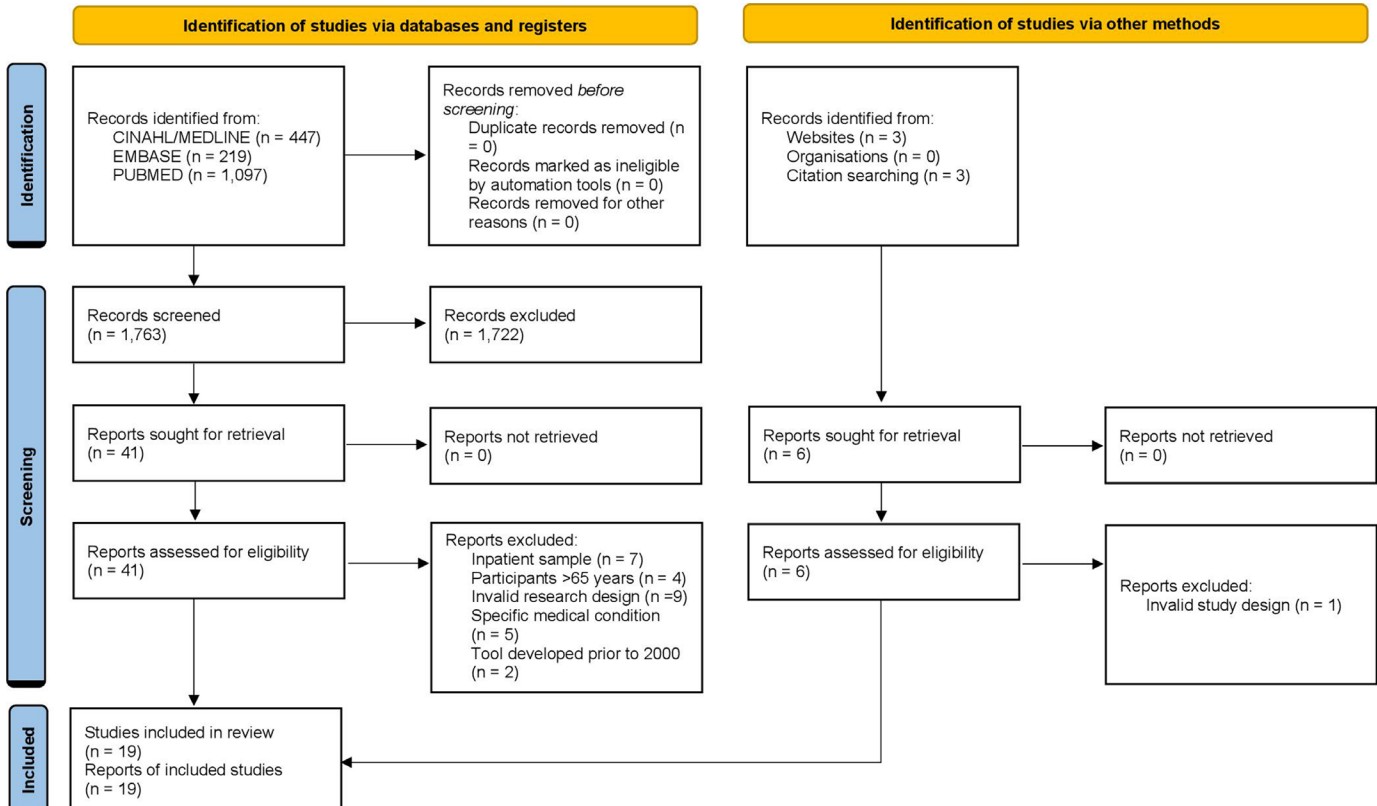

**Figure 1** Study selection.

Fifteen studies included non-pregnant participants. Of these, two studies focused on engagement with mental health services,[34 35] one focused on engagement with child protective services[42] and one solely recruited adolescents.[37] The remaining 11 studies, focused on health engagement in the general adult population with or without chronic conditions, sampled through online or mailed surveys,[30 40] in-hospital outpatient clinics[29 31 32 34 38 39 41] and general practitioner surgeries.[33 36] Three studies used the following conceptual frameworks to design their health engagement instruments: the Patient Health Engagement Scale (PHE-Scale),[32] a conceptual model of outcomes influenced by primary care (Primary Care Outcomes Questionnaire (PCOQ)-24)[36] and therapeutic alliance (Kim Alliance Scale–Revised (KAS-R)).[38]

### Risk of bias in studies

All tools were valid and reliable with at least adequate quality rated on the 11-point scale. Twelve tools were assessed as high quality (scores 9–11), and seven tools were assessed as fair (scores 5–8). The difference in quality related to the extent and adequacy of testing (table 2).

### Results of individual studies

The scales, subscales and items of the 19 included tools were reviewed and mapped against the SHE model constructs (optimal model of care, midwife attributes, best practice principles buy-in) and related elements. No tool measured all components, and one tool[32] measured none of them.

### Health engagement tools for pregnant women

Four tools were designed to measure health engagement in maternity care. Both the Mothers Autonomy in Decision Making (MADM) tool[25] and Mothers on Respect (MoR) Index[27] measured shared decision-making. In addition, the MoR Index[27] measured providers' attributes associated with (dis)respectful care. While the MoR Index[27] measured the unwillingness of pregnant women to ask questions or share concerns with their provider, it did not specifically measure disclosure of risks, which is a significant element of buy-in. The 2014 Quality of Prenatal Care Questionnaire (QPCQ[26]) included six subscales: (1) information sharing, (2) anticipatory guidance, (3) sufficient time, (4) approachability, (5) availability, (6) support and respect. The QPCQ[26] subscales mapped against optimal model of care, provider attributes and best practice principles including health promotion, but did not measure buy-in. The 2004 Prenatal Interpersonal Processes of Care (PIPC[43]) included three dimensions (subscales): (1) communication (elicitation of, and responsiveness to patient; explanations of processes of care; empowerment/self-care); (2) patient-centred decision-making and (3) interpersonal style (friendliness and courteousness; respectfulness/emotional support; lack of perceived discrimination). The PIPC[43] measured provider attributes including being perceived as discriminatory, empowering and empathetic. The PIPC[43] measured elements of best practice including woman-centred care and shared decision-making. While the PIPC[43] was developed with

**Table 2** Characteristics and quality of included tools

| Year Country | Tool and aim | Population | Study design | Quality score |
|---|---|---|---|---|
| **Tools designed for pregnant population accessing maternity care** | | | | |
| 2017[25] Canada | **Mothers Autonomy in Decision Making Scale**. To develop and test a scale to assess women's autonomy and role in decision-making in pregnancy | 1672 women with a single provider during pregnancy | Literature review to generate new items and adapt previous items, modification through survey, focus groups and expert panel review. Four working groups refined content with immigrant and refugee women, formerly incarcerated women, women facing socioeconomic barriers and urban/rural settings. Pilot testing with target population with minor rewording. | 9 |
| 2017[27] Canada | **Mothers on Respect Index**. To assess women's experiences with maternity care, including disrespect and discrimination | | | |
| 2014[26] Canada | **Quality of Prenatal Care Questionnaire**. Develop an instrument to measure the quality of prenatal care | 422 postpartum women (preliminary) 422 postpartum women (final) | Preliminary tool developed following interviews with pregnant women and health providers, review of prenatal care guidelines, assessment of content validity and rating of importance of items. Preliminary testing and exploratory factor analysis to generate and test final version with separate sample of participants. | 10 |
| 2004[43] USA | **Prenatal Interpersonal Processes of Care**. Develop a reliable and valid multidimensional measure of prenatal interpersonal processes of care for use with ethnically diverse women | 363 African American, Latino, Caucasian pregnant women (low income) | Adaptation of the Interpersonal Processes of Care tool. Focus groups were used to test face validity of items prior to telephone survey with the 30-item tool. | 10 |
| **Tools designed for non-pregnant population accessing healthcare** | | | | |
| 2021[28] Canada | **CADICEE**. To co-construct a tool for measuring degree of partnership between patients and healthcare providers | 206 patients and 38 relatives present during consultation | Patients co-constructed tool. Qualitative analysis of patient experience performed by research team. Patient research advisory group involved in entire construction of tool process. Tool tested for construct and convergent validity and internal consistency. | 7 |
| 2019[31] Italy | **Patient-Professional Interaction Questionnaire**. To investigate how patients evaluate provision of patient-centred care by healthcare professionals and psychometrically test a questionnaire | 1139 inpatients and outpatients | A self-assessment of professionals' provision of patient-centred care was adapted into a patient-rated form. The questionnaire structure, reliability, susceptibility to social desirability and associations with other variables were tested. | 8 |

**Table 2** Continued

| Year Country | Tool and aim | Population | Study design | Quality score |
|---|---|---|---|---|
| 2019[30] Germany | **Patient Enablement**. To develop and psychometrically test a German-language survey instrument that measures patient enablement generically and in greater detail than previous instruments | 354 adults registered with the integrated care system | Analysis of key concepts from the literature by a multidisciplinary team-informed item development. Construct and structural validity, and internal consistency were tested. | 9 |
| 2018[36] UK | **Primary Care Outcomes Questionnaire (PCOQ)**. To test the PCOQ which aims to capture a broad range of outcomes relevant to primary care | 602 adult general practitioner patients with and without chronic conditions | Questionnaire developed through use of a conceptual model, interviews with patients and clinicians, expert review through Delphi study and refinement of tool through cognitive interviews prior to psychometric testing. | 10 |
| 2015[32] Italy | **Patient Health Engagement Scale (PHE-Scale)**. To evaluate the psychometric properties of the PHE-Scale and to evaluate the association between PHE-Scale scores and concurrent measures | 382 adults with chronic disease | Items based on previous conceptualisation of patient engagement (the PHE-model). Items based on a systematic analysis of the literature and an extensive qualitative study. Tool was pilot tested and validated using confirmatory factor analysis. | 10 |
| 2015[40] USA | **Altarum Consumer Engagement measure**. To assess an individual's engagement in health and healthcare decisions | 2079 adults through online survey | Item generation through literature review, piloting and refinement of items. Testing of tool through web-based survey. | 9 |
| 2014[37] USA | **Youth Engagement with Health Services**. To create and validate a survey instrument designed to measure youth engagement with health services | 354 ethnically diverse high school students, school health centres | Item generation through literature review including existing validated measures, expert opinion and interviews with adolescents. Instrument pilot tested and refined. | 8 |
| 2010[29] Germany | **Questionnaire on the Quality of Physician-Patient Interaction**. To assess the quality of physician–patient interactions | 147 adults and 19 physicians in outpatient clinic | Item generation from exploratory in-depth interviews with 20 patients. Literature review used to screen for additional items. An expert panel reviewed the final 14-item questionnaire. | 10 |
| 2008[34] Sweden | **Health Promotion Intervention Questionnaire**. To measure patients' subjective experiences of health promotion interventions in mental health services | 135 adults who accessed a mental health service | Item generation informed by qualitative research; cross-section survey used to pilot test tool. | 7 |
| 2008[38] USA | **Kim Alliance Scale–Revised (KAS-R)**. To measure quality of the therapeutic alliance from patient's perspective, including patient empowerment | 601 participants from two outpatient clinics | Conceptual framework developed through literature review and qualitative research. Original 48-item KAS tool piloted as paper-based survey and refined to 30 items. KAS tool further refined to 16 items through exploratory and validation studies. | 10 |

Continued

**Table 2** Continued

| Year Country | Tool and aim | Population | Study design | Quality score |
|---|---|---|---|---|
| 2005[41] USA | **Patient Activation Measure**. To conceptualise and measure 'activation' in patients and consumers | 1515 adults with or without a chronic condition | Mixed-methods study including national expert consensus and patient focus groups to define and identify domains. Pilot testing and refinement prior to national survey. | 10 |
| 2005[42] USA | **Client Engagement in Child Protective Services**. To develop and test a multidimensional measure of client engagement in child welfare services | 287 primary caregivers with an open child protective services case | Five dimensions of engagement based on literature review, interviews with child welfare workers and clients. Three expert panels provided feedback on items, validity and format. Construct validity and internal consistency were tested. | 8 |
| 2004[35] UK | **Engagement Measure (self-report version)**. To develop a reliable self-report measure of engagement based on Hall, Meaden, Smith & Jones 2001 observer-rated measure of engagement | 25 unemployed adults accessing outreach mental health services | Tool developed based on an observer-rated measure of engagement. The *process* of developing the altered self-report measure of engagement not provided. Construct validity, including factor analysis, was not assessed. | 6 |
| 2004[33] Holland | **Patient-Doctor Relationship Questionnaire**. To develop and validate a questionnaire to assess the patient–doctor relationship | 110 general practice patients, 55 epilepsy clinic patients | The Helping Alliance Questionnaire of Luborsky basis for item creation. Pilot testing (n=8) resulted in the additional item. Factor analysis, test/retest reliability and internal consistency were undertaken. | 6 |
| 2001[39] USA | **Facilitation of Patient Involvement Scale** To measure the degree to which patients perceive their physicians actively facilitate or encourage involvement in their own healthcare | 236 adults (pilot) 338 adults (survey) | Item generation through literature review and revision based on expert feedback. Pilot study of 9-item scale using 6-point Likert responses. | 10 |

an ethnically diverse, low-income sample of pregnant women, it does not measure buy-in.

### Health engagement tools for vulnerable people

Two tools were designed for people who were vulnerable due to age (adolescents) or engagement with child protective services. The Youth Engagement in Health Services (YEHS!)[37] survey measured adolescent health engagement across four components: (1) receipt of anticipatory health guidance, (2) experience of care, (3) health access literacy and (4) health self-efficacy. Health topics included in the anticipatory guidance component are relevant to health promotion during pregnancy, and could be easily adapted in future tool development. Yatchmenoff developed a 19-item client-centred health engagement model with four factors: (1) buy-in, (2) receptivity, (3) working relationship and (4) mistrust.[42] The factors, and items, in this tool are relevant to buy-in and receptivity of vulnerable pregnant populations and the midwife–woman relationship.

### General health activation tools

The Patient Activation Measure (PAM-13)[41] was the first tool to define the construct of patient activation and is the most widely used measure of patient engagement. The 13 items focus on the patient's perceived knowledge, confidence and ability to manage their health condition.[41] While commonly used, the tool has limited generalisability to vulnerable populations because of readability for people with low health literacy.[44] Furthermore, because activation is constructed solely in terms of *confidence* and *ability*, this obfuscates significant barriers

to health engagement that occur through social determinants of health and poor patient–provider relationships.

The Patient Enablement (PEN-13) Questionnaire (German version) focuses specifically on measuring skills patients without chronic conditions had to manage their healthcare.[30] Like the PAM-13,[41] this tool measures an aspect of patient engagement without the context of the patient–provider relationship, which limits usefulness for modification for a maternity tool.

The PHE-Scale[32] focuses on patients who receive a health diagnosis, and measures items that occur prior to, and predict, the individual's ability to engage in healthcare. For example, *When I think about my disease: I feel blackout, I am in alarm, I am aware, I feel positive.*[32] For both reasons, the PHE-Scale[32] is not relevant to maternity health engagement.

The Altarum Consumer Engagement (ACE) measure[40] included 20 items evenly divided across four subscales: (1) commitment, (2) informed choice, (3) navigation and (4) ownership. The ACE[40] measures individuals' confidence to navigate the healthcare system and take responsibility for their health. Only one item refers to interaction with a health professional (ie, *I feel comfortable talking to my doctor about my health*).[40]

### Patient–provider relationship tools

Six tools focused on health engagement through the lens of the patient–provider relationship. The KAS-R[38] was developed to measure *therapeutic alliance* in healthcare. The KAS-R[38] included 16 items evenly divided across four subscales: (1) collaboration, (2) integration, (3) empowerment and (4) communication. Items which measured mutual goals and respect are useful to health engagement for pregnant women. The wording of some items (eg, *I am allowed in the decision-making process*)[38] reflects that this tool was developed over 15 years ago, prior to significant progress in the area of shared decision-making. More recently published tools, like the MADM tool,[25] have more appropriate wording and description of decision-making in maternity care.

The Patient-Doctor Relationship Questionnaire (PDRQ-9)[33] contained nine short and general, and potentially vague, statements about satisfaction with relationship with the primary care provider (eg, *My primary care provider understands me*). In comparison, the Questionnaire on Quality of Physician-Patient Interaction (QQPPI)[29] contained 14 statements which were more specific (eg, *The physician seemed genuinely interested in my problems*). Likewise, the Facilitation of Patient Involvement (FPI) Scale[39] asked patients to rate how often their physician does nine things. For example, *My doctor explains all the treatment options to me so that I can make an informed choice.*[39] The Patient-Professional Interaction Questionnaire (PPIQ)[31] included 16 items, of which several measured whether the professional was interested in (eg, *what I know about my disease/prognosis, what I want from care*). The Health Promotion Intervention Questionnaire (HPIQ)[34] measures patient experiences of health promotion as received through their relationship with a key worker. The HPIQ[34] consists of 19 items across four factors: alliance, empowerment, educational support and practical support. The items are commonly superficially worded (eg, *Key worker treats me in a friendly way and often smiles*), which does not measure the therapeutic nature of the patient–provider relationship.

### Comprehensive health engagement tools

Three tools measured both patient–provider relationship and components of patient activation. The CADICEE tool[28] comprehensively addresses the complex components of health engagement using 24 items across seven dimensions: (1) relationship of **C**onfidence or trust between patients and healthcare providers, (2) patient **A**utonomy, (3) patient participation in **D**ecisions related to care, (4) shared **I**nformation on patient health status and care, (5) patient personal **C**ontext, (6) **E**mpathy and (7) recognition of **E**xpertise.[28]

The PCOQ-24[36] has 24 items across four dimensions: (1) health and well-being, (2) health knowledge and self-care, (3) confidence in health provision and (4) confidence in health plan. Items ask about the skills and attributes of *the doctors or nurses you usually see* as well as patient knowledge and ability to self-manage their health condition.[36]

The Engagement Measure (EM) (client version)[35] has six key areas: (1) appointment keeping, (2) client–key worker relationship, (3) communication/openness with key workers, (4) usefulness of treatment, (5) involvement with treatment and (6) taking medication. The professional's attributes were not described (*eg, How well do you get on with _______?*), which makes items so general that their value is limited.

### Synthesis of results

The results from all studies were synthesised through mapping against SHE model constructs of model of care; health professional's skills and attributes; best practice and buy-in.

### Model of care

The availability and flexibility of the health professional to meet the individual's needs were largely measured in terms of having sufficient time and being contactable. For example, the PIPC item: *How often did providers give you enough time to say what you thought was important?*[43] The QPCQ[26] differentiated between being able to contact *someone* versus *my provider* when concerns arose. Four general health engagement tools (QQPPI,[29] PPIQ,[31] PDRQ-9,[33] YEHS![37]) included items about having *enough time* with a health professional. Only one general tool (PDRQ-9[33]) measured accessibility: *I find my primary care provider easily accessible.*

### Health professionals' skills and attributes

Most tools include subscales or items that measured whether the individual perceived their provider as empowering, trustworthy or empathetic, or provider actions

that could be interpreted to demonstrate these qualities. Of the four maternity tools, two tools included the terms 'doctor or midwife' and asked respondents to indicate which healthcare provider their responses referred to.[25 27] On the other hand, the QPCQ referred generally to a *prenatal care provider,*[26] and the PIPC measured experiences about *doctors, nurses and other providers.*[43]

### Empowering

An empowering health professional refers to someone who uses communication skills to effectively share information with pregnant women. Any items that were specifically about how the health professional facilitated shared decision-making were mapped against that element of best practice. Two maternity tools measured attributes related to whether the health professional was empowering. For example, the QPCQ item: *I fully understood the reasons for blood work and other tests that my prenatal care provider(s) ordered for me*[26] and the PIPC item: *How often did providers tell you what they were doing as they gave you a physical examination?*[43]

Most non-maternity tools included at least one item that measured whether the health professional acted in ways that were empowering (QQPPI,[29] PPIQ,[31] PCOQ,[36] CADICEE,[28] PAM-13,[41] KAS-R,[38] FPI,[39] PEN-13,[30] YEHS![37]). The KAS-R[38] included an empowerment subscale with items such as: *I have an active partnership with my provider,* whereas the FPI[39] included a negatively worded item: *My doctor discourages my questions.*

### Trustworthy

Three maternity tools (QPCQ,[26] PIPC,[43] MoR[27]) measured outcomes related to perceptions of the health professional being trustworthy, including perceived discrimination, support and respect. However, only two tools specifically measured racism. For example, the PIPC item: *How often did you feel discriminated against because of your race or ethnicity?*[43] On the other hand, the MoR included multiple items related to disrespectful care including *When I had my baby I felt that I was treated poorly by my (midwife, doctor): Because of my race, ethnicity, cultural background or language.*[27]

One-third of the non-maternity tools (YEHS!,[37] Client Engagement in Child Protective Services (CECPS),[42] CADICEE,[28] PCOQ,[36] PDRQ-9[33]) measured patient perceptions of patient–provider relationship in terms of trust. For example, the YEHS! item: *I have a safe and trusting relationship with at least one doctor or healthcare provider.*[37] On the other hand, the PCOQ-24[36] included a more generic question regarding doctors and nurses in the general practice: *You can trust them.* This item does not speak to a patient–provider relationship, nor continuity of care with a specific provider, so it of limited utility.

### Empathetic

Both the QPCQ[26] and PIPC[43] have subscales related to empathy including demonstrating interest in the woman as a person and communicating emotional support. For example, the QPCQ item: *My prenatal care provider was interested in how my pregnancy was affecting my life*[26] and the PIPC item: *How often did providers help you feel less worried about your pregnancy?*[43]

About half of the general tools (YEHS!,[37] QQPPI,[29] CADICEE,[28] PPIQ,[31] KAS-R,[38] HPIQ[34]) measured patient–provider relationship in terms of empathy. Similarly, to the PIPC item, a QQPPI item measured the ability of the provider to reassure the patient: *The physician did all he/she could to put me at ease,*[29] whereas some tools more explicitly measured the emotional element of the patient–provider relationship. For example, the PPIQ item: *He/she was able to put him/herself in 'my shoes'.*[31]

### Best practice principles

In the SHE model, health promotion, woman-centred care and shared decision-making were captured as a construct titled *philosophical commitments,* in other words, best practice principles.

### Health promotion

Health promotion in maternity care includes discussion and recommendations around diet, exercise, emotional health, interpersonal relationships, social support, and harm minimisation for smoking, drugs and alcohol.[20] Two included maternity tools measured constructs directly related to promotion of healthy behaviours. A QPCQ subscale measured whether women were provided with education about components of health including diet, exercise, alcohol, emotional health; for example: *I was given adequate information about depression in pregnancy.*[26] In contrast, a PIPC subscale asked more broadly about whether pregnant women had received lifestyle advice.[43] The PIPC also included an item about the health professional's role in promoting the woman to engage in healthy behaviours: *How often did providers tell you what you could do to take care of yourself and your pregnancy at home?*[43]

Few non-maternity tools included items around health promotion. However, both PAM-13[41] and ACE[40] included several items around patient responsibility in self-care and healthy behaviours. On the other hand, the HPIQ, designed for participants with mental health concerns, includes the item: *My key worker informs me about what I need in order to feel better.*[34] The YEHS! was the most relevant because it provided a list of specific educational topics beyond diet and exercise, relevant to vulnerable populations:

> In the last 12 months, did a doctor or other healthcare provider talk with you about the following: weight, healthy eating or diet, physical activity or exercise, your emotions or moods, how you deal with stress, sleep, sexual risk reduction (sexually transmitted diseases).[37]

### Woman-centred/individualised care

Most maternity tools measured whether the individual felt care was individualised to their specific values, preferences

and desires (QPCQ,[26] MoR,[27] PIPC[43]). For example, the QPCQ item: *I was supported by my prenatal care provider(s) in doing what I felt was right for me.*[26] Only three non-maternity tools assessed whether the provider communicated interest in what the patient knew and wanted from their care (QQPPI,[29] CADICEE,[28] PPIQ[31]). For example, the CADICEE tool measured a further element of individualisation around patient expertise about themselves and their health: *Did you feel that this professional recognised your expertise and took it into account?*.[28]

### Shared decision-making

Shared decision-making was measured in all four maternity tools. The most comprehensive measures were included in the MADM tool[25] and MoR Index,[27] which were co-designed with vulnerable pregnant women. Example items include: *My doctor or midwife explained the advantages and disadvantages of the maternity care options* (MADM[25]) and *Overall, while making decision during my pregnancy, I felt coerced into accepting the options my (midwife, doctor) recommended* (MoR[27]). About half of the non-maternity tools included items that measured shared decision-making (QQPPI,[29] CADICEE,[28] PPIQ,[31] KAS-R,[38] FPI,[39] HPIQ[34]). For example, the QQPPI item: *The physician and I made treatment decisions together.*[29]

### Buy-in

Buy-in was not measured by the included maternity tools, whereas each element (self-care, hope/belief, disclosure of risk and accepting help) was measured by some of the non-maternity tools.

### Self-care

Several non-maternity tools (PAM-13,[41] ACE,[40] PCOQ,[36] PEN-13[30]) measured the individuals' confidence and willingness to manage their health including knowing when and how to access help, following through on plans and goals, maintaining a healthy lifestyle and preventing deterioration of their health. For example, the PAM-13 item *I am confident that I can take actions that will help prevent or minimise some symptoms or problems associated with my health condition.*[41] These items were generally worded around a diagnosed health condition, which does not translate easily to pregnancy. The PCOQ-24[36] included a more general question around self-care and health: *Thinking about your level of knowledge, how much do you know how best to look after yourself and stay healthy?*

### Hope and belief

An individual's mindset, their interpretation of whether healthcare is 'worth it' or not, is a motivating factor in health engagement. This element was measured by one-third of the non-maternity tools (CECPS,[42] PCOQ,[36] PPIQ,[31] EM,[35] HPIQ[34]), commonly in a single item. The CECPS[42] included five items that measured mutual goals and optimism for a better future through engagement with child protective services. For example, *Working with _______ has given me more hope about how my life is going to go in the future.*[42]

### Disclosure

The element of disclosure of risk factors (eg, family violence) was not specifically measured by any of the 19 included tools. Of the four maternity tools, only the MoR Index[27] included three items regarding whether women perceived they were held back from asking questions or discussing concerns. About half of the non-maternity tools (YEHS!, CECPS,[42] QQPPI,[29] CADICEE,[28] EM, PAM-13,[41] PEN-13[30]) included items related to disclosure. For example, whether individuals could share problems or concerns without being directly asked by the provider (*I have no difficulty in telling my doctor about my concerns or fears, even if he or she does not address them directly*—PEN-13[30]). On the other hand, the CECPS includes an item about non-disclosure (*Anything I say they're going to turn it around and make me look bad*).[42]

### Accepting help, referral, treatment

Acceptance of help, including agreeing to referrals and following up with tests, community resources and support from other providers, was not measured. Some maternity tools measured whether information was provided about screening tests.[43] Providing an explanation is empowering and may increase the likelihood of women doing the test, but it does not measure buy-in. Agreeing to treatment, going along with treatment and wanting help were measured by non-maternity tools. One-third of the non-maternity tools (YEHS!,[37] CECPS,[42] PCOQ,[36] EM,[35] PAM-13[41]) measured willingness to accept or act on help offered by the care provider. For example, *When I make a plan with a doctor or other healthcare provider, I can follow through on the plan at home* in the YEHS![37] The PCOQ-24 includes items with greater specificity around follow through, for example: *How much of your doctors' or nurses' advice are you following on: your medication or treatment, leading a healthy lifestyle?*[36]

### DISCUSSION

This is the first systematic review to evaluate the characteristics, quality and sufficiency of tools to measure health engagement in vulnerable pregnant populations. There is level 1 evidence that midwifery-led care delivers a 24% reduction in preterm birth.[23] For women at highest risk, like First Nations, there is emerging evidence that the rate of reduction is 38%.[45] No other clinical intervention delivers a benefit of this magnitude to prevent preterm birth. Yet, how midwifery-led care works to reduce preterm birth is not clear. The SHE model hypothesises that midwifery-led care works, through the mechanism of health engagement, to impact modifiable, social predictors of preterm birth.[20] This hypothesis requires testing through measuring health engagement in vulnerable women who access midwifery-led versus other models of care. The results of this systematic review indicate that a tool needs to be developed, which is psychometrically validated to measure health engagement for vulnerable pregnant women. Twelve tools were high quality and validated

for use with generally healthy adolescents and adults in high-income healthcare settings. The SHE framework was useful to map constructs to potential scales, subscales and items. Buy-in was not measured in the included maternity tools, and only partially measured in some of the general patient engagement tools. Specifically, disclosure and accepting help were rarely measured.

The review was limited to studies with abstracts available in English published from 2000 onwards. There may be studies in different languages or published prior to 2000, which were not located or excluded from review. While we did review tools used with other populations, our focus on concepts of care relevant to pregnant women may have been limiting. However, registration of the review protocol confirmed the relative lack of attention to health engagement generally and with pregnant women specifically. The inclusion of tools that were psychometrically tested with individuals with chronic disease (eg, mental health, diabetes) was a poor fit for vulnerable pregnant women who are generally physically healthy. The breadth of included tools may therefore be considered a limitation. Although, application of the SHE model as a framework for the review ensured that a range of relevant concepts were considered, two elements (health literacy and cultural safety) were identified as missing. Inclusion of these elements from the outset may have influenced the search terms and outcomes. Furthermore, the SHE model has been tested only with adolescent mothers.

## Health literacy
Elements of health literacy were broadly measured including how well participants could understand instructions or explanations provided to them by care providers (KAS-R,[38] QQPPI[29]), or how users sought and used information (ACE[40]). Interestingly, the provider's knowledge and skills around maternal health literacy were not measured in the included maternity engagement tools. A national survey reported that most Australian midwives have not received education about health literacy, do not formally assess it, and either never or only sometimes use techniques to facilitate it.[46] There are currently no rapid measures of maternal health literacy available.[47]

## Cultural safety
Racism undermines equitable access to services and predicts poor maternal and perinatal outcomes.[48] Internationally, First Nations women experience racism, prejudice and discrimination, and lack of respect for cultural practices in maternity care.[49 50] Indeed, social risk factors are exacerbated by healthcare that is perceived as racist or culturally unsafe because it contributes to reduced adherence to health advice or complete disengagement from services.[51] Women who do not wish to attend, or are prevented from attending, antenatal care are less likely to receive advice, screening and reassurance about the progress of pregnancy. Beyond attendance, cultural safety is central to women's decision to disclose risk factors, like domestic violence, to their care providers.[52] Perceived lack of cultural safety through hospital-based, fragmented care is a barrier to attendance, whereas culturally safe, midwifery continuity of care in the community is facilitative.[53] A recent study of the impact of culturally safe maternity care concluded that synergistic engagement is the mechanism by which these innovative services effectively improve maternal and perinatal outcomes for First Nations women in Australia.[54] However, in the absence of a tool validated to measure SHE constructs, this explanation remains theoretical.

## Antenatal attendance
The term *engagement* is commonly used as a synonym for *attendance* in the research literature. For example, a recent scoping review on improving antenatal engagement for Aboriginal women used measures of non-attendance, less attendance and late attendance.[55] While antenatal attendance is important, without *buy-in* to maternity care and resultant modification of social risk factors, its efficacy is limited. The ability to determine appropriate medical and/or psychosocial intervention is predicated on women feeling safe to disclose risks to their maternity care providers.[56] Future studies of maternity care engagement should include specific measures that can assess not only the quantity of engagement (attendance) but quality.

## Relationship-based care
Testing of health-promoting interventions, delivered in the absence of relationship-based care, has been proposed to improve perinatal outcomes.[57] While health promotion is an important part of quality maternity care, the provision of motivational counselling around social risk factors does not necessarily translate into women acting on advice. A trusting provider–woman relationship is foundational to pregnant women's willingness and ability to engage.[20] Women randomised to receive midwifery continuity, versus fragmented care, characterised their midwife as *empowering* and *endorphic* (makes me feel safe, loved, relaxed), compared with simply informative, competent, kind.[58] We recommend interventions designed to promote healthy behaviours during pregnancy are embedded in relationship-based models of care to bolster their efficacy.

## Implications for practice and future research
Maternal health engagement is crucial to improving perinatal outcomes for vulnerable groups. Models of maternity care can be designed to enhance or hinder engagement. Therefore, routine measurement of health engagement is an important activity to monitor and improve the quality of maternity care for vulnerable pregnant women. While 12 tools were deemed to be high quality, no single tool addressed all relevant constructs. A new assessment tool is required that addresses all the relevant constructs, developed for and psychometrically assessed in the target group.

**Contributors** JA conceived and designed the analysis, collected the data, analysed the results and wrote the paper. DC conceived and design the analysis, performed quality appraisal of included articles and revised the paper. KM performed quality appraisal of included articles and revised the paper. JG conceived and designed the analysis and revised the paper. Guarantor, JA.

**Funding** The authors have not declared a specific grant for this research from any funding agency in the public, commercial or not-for-profit sectors.

**Competing interests** None declared.

**Patient and public involvement** Patients and/or the public were not involved in the design, or conduct, or reporting, or dissemination plans of this research.

**Patient consent for publication** Not required.

**Ethics approval** Ethical approval was not required for this review.

**Provenance and peer review** Not commissioned; externally peer reviewed.

**Data availability statement** Data sharing not applicable as no datasets generated and/or analysed for this study.

**ORCID iDs**
Jyai Allen http://orcid.org/0000-0003-3196-0883
Debra K Creedy http://orcid.org/0000-0002-3046-4143
Jenny Gamble http://orcid.org/0000-0002-3749-3473

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
