## [Reviewer comments · BMJ Open]

ARTICLE DETAILS

TITLE (PROVISIONAL)	Health engagement: A systematic review of tools modifiable for use with vulnerable pregnant women
AUTHORS	Allen, Jyai; Creedy, Debra; Mills, Kyly; Gamble, Jenny

VERSION 1 – REVIEW

REVIEWER	Marianne Rasmussen Copenhagen University Hospital Bispebjerg-Frederiksberg, The Parker Institute
REVIEW RETURNED	12-Aug-2022

GENERAL COMMENTS	I would like to thank the editors of BMJ Open, for letting me review this excellent manuscript, and I give my compliments to the authors for conducting this thorough and very well written paper on the important subject of identifying appropriate tools to assess health engagement in vulnerable pregnant women. Such tools are important in the assessment of the health engagement of vulnerable pregnant women who are in risk of poorer maternal health outcomes, to enable and initiate appropriate measures to promote healthier maternal health outcomes in this target group. The authors have applied a thorough methodology and used the appropriate checklists in identifying appropriate measurement tools for vulnerable pregnant women, and although only four out of the 19 tools identified, were developed for, and tested within a population of pregnant women, and only two out of these were tested in the target population of vulnerable pregnant women, all identified tools were mapped against the SHE model constructs, to ensure the utility of tools in that the tools included important constructs relevant for health engagement in vulnerable pregnant women. There are only a few comments to address: In the result section, under study selection, page 26 line 18, it is stated that two authors read 47 full-text articles, but according to figure 1, it was 41 articles. Please correct. The 19 identified health measurement tools are rated according to the COSMIN Risk of Bias checklist, and given a quality score according to their psychometric properties, displayed in table 1. The authors do acknowledge that a minority of the tools identified are tested and validated in the target group, Still, many of the tools get a high psychometric quality score (table 1), even when many of the tools are developed and tested for patients with illnesses and chronic conditions. This is a limitation to the study, and although this
---

	problem is addressed briefly in the discussion section on in the second paragraph, line 13 to 15, this limitation should be further elaborated on, both in the discussion section and in the implication for future research section. In conclusion it should be stated that a new assessment tool that addresses all the relevant constructs and are developed for, and psychometrically assessed in the target group, is called for. Overall, this well-written manuscript represents a valuable contribution to the work of identifying and target interventions for pregnant women in risk of poorer maternal health outcomes, and it is highly recommended for publication.
--	---

REVIEWER	Tanvir Turin University of Calgary, Department of Family Medicine, Cumming School of Medicine
REVIEW RETURNED	09-Dec-2022

GENERAL COMMENTS	The authors conducted systematic research to identify existing primary literature on health engagement tools for pregnant women. It is decently-written. The authors have done a decent review of existing studies that used health engagement frameworks to evaluate which tools, or components, could be modified for use with vulnerable pregnant populations. Few things need to be addressed - - Comment 1 ----- * Abstract. The objective needs to be clarified in the abstract according to the method section of the manuscript. Authors could better include more specific and significant information, like which type of systematic review they conducted following which synthesis method, the number of studies from the initial search to the final selection etc. Comment 2 ----- Use the term “vulnerable pregnant population” or “vulnerable population” consistently across the manuscript. Page 3, line 48, The vulnerable pregnant population was used. Again on page 4, line 52, The term ‘adolescent or adults’ was used in the inclusion criteria. Please clarify the inclusion criteria with the keywords of the manuscript. Comment 3 ----- Page 5, line 11 A table containing all the searched keywords and names of the databases will be clearer and more helpful for the presentation. Comment 4 ----- Page 6, line 4
--

Authors should elaborate on the theoretical framework more, especially how they design search terms from the theoretical framework needs to be explained.

Line 44-46

“Application of the SHE model informed the selection of search terms and interpretation of Findings”- this line should be explained clearly.

Comment 5

Page 6, line 17

In the “study selection paragraph”, a comprehensive number of the studies, from initial searches to the final selection, could better portray how many studies were found from the initial search and how many were finally selected.

Comment 6

On page 8, Table 1, the authors considered studies on pregnant and non-pregnant women. But, on abstract Page 1, line 7, the objective is to examine available health engagement tools for the vulnerable pregnant population. Please clearly write the objectives across the manuscript.

Comment 7

Page 10 Table 1

References (author's name and year) should be added to the selected studies in the first column of the table.

Comment 8

Page 27, line 6

In the PRISMA flow chart “Identification of studies via other methods”, I didn’t see any description of the other methods in the manuscript: how these methods were executed and the rationale for doing these other methods. Please explain other methods used in this review in the method section.

Comment 9

Page 27, lines 41 &41

In the PRISMA flowchart: the total number of included studies and reports was 38, but the authors finally selected 19 papers. Also, the number of reports needs to be consistent across the flowchart. Please correct the number of studies.

Comment 10

Page 22, lines 5-9

“Development and testing of a tool that addresses these concepts within a revised Synergistic Health Engagement model is required.” The authors should explain more why this model is required.

VERSION 1 – AUTHOR RESPONSE

Reviewer: 1	
Overall, this well-written manuscript represents a valuable contribution to the work of identifying and target interventions for pregnant women in risk of poorer maternal health outcomes, and it is highly recommended for publication.	We appreciate the reviewer's assessment of this paper as worthy of publication.
1. In the result section, under study selection, page 26 line 18, it is stated that two authors read 47 full-text articles, but according to figure 1, it was 41 articles. Please correct.	Thank-you for noting this error. Figure 1 is correct, and the text has been revised to reflect this as 41 articles. Change highlighted.
2. The 19 identified health measurement tools are rated according to the COSMIN Risk of Bias checklist and given a quality score according to their psychometric properties, displayed in table 1. The authors do acknowledge that a minority of the tools identified are tested and validated in the target group, Still, many of the tools get a high psychometric quality score (table 1), even when many of the tools are developed and tested for patients with illnesses and chronic conditions. This is a limitation to the study, and although this problem is addressed briefly in the discussion section on in the second paragraph, line 13 to 15, this limitation should be further elaborated on, both in the discussion section and in the implication for future research section.	This has been more clearly addressed as a limitation in Discussion section. Change highlighted.
3. In conclusion it should be stated that a new assessment tool that addresses all the relevant constructs and are developed for, and psychometrically assessed in the target group, is called for.	The Conclusion in the Abstract has been revised accordingly. Changes highlighted.
Reviewer: 2	
Comment 1 ----- * Abstract. The objective needs to be clarified in the abstract according to the method section of the manuscript. Authors could better include more specific and significant information, like which type of systematic review they conducted following which synthesis method, the number of studies from the initial search to the final selection etc.	The Abstract meets the requirements of the PRISMA 2020 Abstract checklist for systematic reviews (i.e., number of included studies only). No change.
Comment 2 -----	While we understand it is important to use a consistent term for the same concept, the

Use the term “vulnerable pregnant population” or “vulnerable population” consistently across the manuscript. Page 3, line 48, The vulnerable pregnant population was used. Again on page 4, line 52, The term ‘adolescent or adults’ was used in the inclusion criteria. Please clarify the inclusion criteria with the keywords of the manuscript.	terms ‘vulnerable pregnant population’ and ‘vulnerable population’ are not synonyms. Vulnerable populations were defined as people with low health literacy, socio-economically disadvantaged, adolescents (and tools were included that had been tested in those populations). Vulnerable pregnant populations were defined as women who have those characteristics and are also pregnant. No change.
Comment 3 ----- Page 5, line 11 A table containing all the searched keywords and names of the databases will be clearer and more helpful for the presentation.	Thank you, yes this is much clear now that it has been revised into a Table. Change highlighted.
Comment 4 ----- Page 6, line 4 Authors should elaborate on the theoretical framework more, especially how they design search terms from the theoretical framework needs to be explained. Line 44-46 “Application of the SHE model informed the selection of search terms and interpretation of Findings”- this line should be explained clearly.	Thank you for giving us the opportunity to clarify this, because the SHE model did not design the search terms but did provide a framework for analysis of the items in the included tools. Change highlighted.
Comment 5 ----- Page 6, line 17 In the “study selection paragraph”, a comprehensive number of the studies, from initial searches to the final selection, could better portray how many studies were found from the initial search and how many were finally selected.	We have followed BMJ Open author guidelines which stipulate that “Tables should be self-explanatory and the data they contain must not be duplicated in the text or figures.” Therefore, numbers reported in Figure 1 are not duplicated in text. No change.
Comment 6 ----- On page 8, Table 1, the authors considered studies on pregnant and non-pregnant women. But, on abstract Page 1, line 7, the objective is to examine available health engagement tools for the vulnerable pregnant population. Please clearly write the objectives across the manuscript.	Objective: to examine available health engagement tools suitable to, or modifiable for, vulnerable pregnant populations. The Abstract objective does not state the study was limited to health engagement tools designed for the vulnerable pregnant population. Instead, it was to find tools in the broader health literature that may be suitable or modifiable for the development of a health

	engagement tool for vulnerable pregnant populations. The Methods are therefore consistent with the stated objective. However, the title has been revised for consistency between title, objective, and methods. Change highlighted.
Comment 7 ----- Page 10 Table 1 References (author's name and year) should be added to the selected studies in the first column of the table.	Agreed, however, the author's name was removed in order to meet BMJ Open's strict submission criteria for no more than 2-pages for a Table. No change.
Comment 8 ----- Page 27, line 6 In the PRISMA flow chart "Identification of studies via other methods", I didn't see any description of the other methods in the manuscript: how these methods were executed and the rationale for doing these other methods. Please explain other methods used in this review in the method section.	Thank you, you have raised an important point here as we have not been clear. There was not a separate search of websites – instead the authors were aware of 3 tools (MADM, MORI, YEHS!) that were relevant to the review yet not identified through the search strategy of medical databases. These 3 tools were explicitly searched for and found on websites. This explanation has now been provided in the method section. Change highlighted.
Comment 9 ----- Page 27, lines 41 &41 In the PRISMA flowchart: the total number of included studies and reports was 38, but the authors finally selected 19 papers. Also, the number of reports needs to be consistent across the flowchart. Please correct the number of studies.	Consistency between text and Figure 1 has been cross-checked and clarified. Change highlighted.
Comment 10 ----- Page 22, lines 5-9 "Development and testing of a tool that addresses these concepts within a revised Synergistic Health Engagement model is required." The authors should explain more why this model is required.	This is an excellent question and speaks to significance which had not been clearly addressed. The Discussion has been revised to address this. Change highlighted.

VERSION 2 – REVIEW

REVIEWER	Tanvir Turin University of Calgary, Department of Family Medicine, Cumming School of Medicine
REVIEW RETURNED	12-Feb-2023
GENERAL COMMENTS	The reviewer completed the checklist but made no further comments.